# YASTN: Yet another symmetric tensor networks; A Python library for abelian symmetric tensor network calculations.

**Marek M. Rams[1*], Gabriela Wójtowicz[1,2†], Aritra Sinha[1,3‡] and Juraj Hasik[4,5°]**

**1** Jagiellonian University, Institute of Theoretical Physics, Łojasiewicza 11, 30-348 Kraków, Poland
**2** Institut für Theoretische Physik und IQST, Albert-Einstein-Allee 11, Universität Ulm, D-89081 Ulm, Germany
**3** Max Planck Institute for the Physics of Complex Systems, Nöthnitzer Strasse 38, Dresden 01187, Germany
**4** Institute for Theoretical Physics and Delta Institute for Theoretical Physics, University of Amsterdam, Science Park 904, 1098 XH Amsterdam, The Netherlands
**5** Department of Physics, University of Zurich, Winterthurerstrasse 190, 8057 Zurich, Switzerland

* marek.rams@uj.edu.pl , † gabriela.wojtowicz@uni-ulm.de , ‡ asinha@pks.mpg.de , °
juraj.hasik@physik.uzh.ch

## Abstract

**We present an open-source tensor network Python library for quantum many-body simulations. At its core is an abelian-symmetric tensor, implemented as a sparse block structure managed by a logical layer on top of a dense multidimensional array backend. This serves as the basis for higher-level tensor network algorithms operating on matrix product states and projected entangled pair states. An appropriate backend, such as PyTorch, gives direct access to automatic differentiation (AD) for cost-function gradient calculations and execution on GPU and other supported accelerators. We show the library performance in simulations with infinite projected entangled-pair states, such as finding the ground states with AD and simulating thermal states of the Hubbard model via imaginary time evolution. For these challenging examples, we identify and quantify sources of the numerical advantage exploited by the symmetric-tensor implementation.**

# 1 Introduction

Full numerical treatment of quantum-mechanical systems is generally prohibitively expensive, due to the exponential growth of the Hilbert space size with the number of interacting degrees of freedom. *Tensor network* (TN) techniques allow efficient representation and manipulation of the states of such large quantum systems [1–3]. The *density matrix renormalization group* (DMRG) introduced by White [4, 5] and its modern reformulation in terms of *matrix product states* [6–10] (MPS), a one-dimensional (1D) tensor network, is a prime example of TN capabilities. Since their inception, MPS quickly became a reference method for addressing ground states in 1D, and were soon followed by extensions to excited states, time evolution, and open systems, forming a comprehensive framework.

The descriptive power of TNs generalizes to higher-dimensional models. The MPS, despite its intrinsic 1D geometry, can be readily applied to systems in higher dimensions by imposing linear ordering of lattice sites. Two-dimensional (2D) systems are often limited to finite cylinders that are mapped to the MPS ansatz by imposing ordering that winds around the cylinder circumference; see Fig. 1(c). More natural TN geometry for 2D states is assured by the *projected entangled-pair states* (PEPS) [11,12], see Fig. 1(e), and the similar for 3D states [13,14].

The TN ansätze in Fig. 1 provide a state-of-the-art numerical approach to strongly correlated systems of condensed matter. The computational complexity of MPS typically scales as $\mathcal{O}(D^3)$, and PEPS algorithms often reach $\mathcal{O}(D^{12})$ scaling, where bond dimension $D$ governs the size of the tensors and the overall precision of the TN approximations. Although the scaling of PEPS seems less favorable, it is important to note that the bond dimension encodes correlations between sites. Imposing winding, column-by-column ordering for the MPS on a cylinder stretches the correlations between columns and leads to long-range correlations across the MPS. The PEPS, on the other hand, already possesses natural geometry for nearest-neighbor correlations in 2D. As a result, the PEPS can reach comparable or better precision even at low bond dimensions once the cylinder width within the MPS approach exceeds a few sites.

The most effective way to mitigate computational complexity is to take advantage of the symmetries present in physical systems. Two principal types of symmetries to consider are spatial and internal symmetries. Tensor networks can be formulated directly in the thermodynamic limit by an infinitely repeated pattern (a unit cell) of tensors, hence realizing translation symmetry. These are infinite-MPS (iMPS), also known as uniform MPS [15] in 1D and infinite-PEPS (iPEPS) [16] in 2D. The computational complexity of the iMPS/iPEPS algorithms scales linearly with the size of the unit cell.

For internal symmetries $U|\Psi\rangle = |\Psi\rangle$, we consider their common form of global symmetries, i.e., when $U = \otimes_i u_i$ with the same unitaries $u_i$ acting on each lattice site. These can be both abelian (e.g., particle conservation) or non-abelian (e.g., SU(2)-spin). Crucially, such global symmetries can be implemented in TNs locally by requiring individual tensors to transform covariantly under the action $u$ of the symmetry group [17–21]. These symmetric tensors take

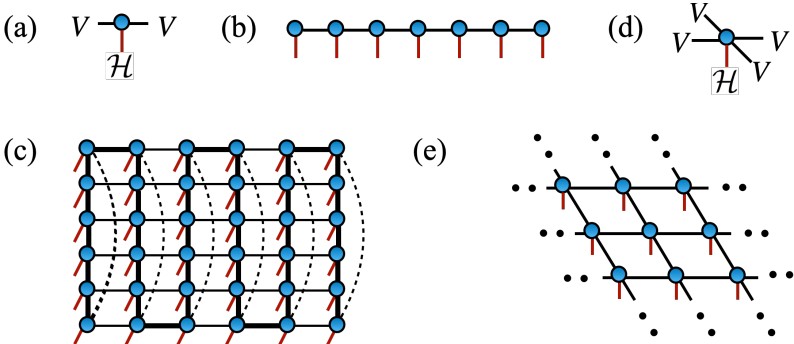

Figure 1: **Tensor networks.** Diagrams depict (a) a rank-3 tensor of the MPS with left and right virtual spaces $V$ and physical space $\mathcal{H}$, (b) an MPS ansatz, (c) a MPS winding on a finite width-6 cylinder, (d) a rank-5 tensor of the PEPS, and (e) an infinite-PEPS ansatz.

block-spare form, with original dense virtual spaces $V$ of bond dimension $D$ split into a direct sum of blocks $V = \oplus_r V_r$ with dimensions $\{D_1, \ldots, D_r\}$ each associated with irreducible representation $r$ of the symmetry group considered. Block sparsity substantially reduces computational complexity, permitting large $D$ simulations, in particular, for (i)PEPS algorithms.

Here, we introduce the *Yet Another Symmetric Tensor Network* (YASTN) library [22]. YASTN is an open-source Python library with abelian-symmetric tensor as a basic type and associated linear algebra operations on such tensors. The implementation enables *automatic differentiation* (AD) via appropriate dense linear algebra backends, allowing convenient variational optimization of TNs. This is particularly important for iPEPS [23–25], where no alternative direct energy minimization algorithms are known. This is in contrast to (i)MPS where the DMRG provides efficient and robust optimization. YASTN thus joins a continually growing collection of tensor network software with various degree of support for symmetries and automatic differentiation, such as ITensor [26], TenPy [27], Block2 [28], Quantum TEA [29], TensorNetwork [30], Cytnx [31], TeNes [32], TensorKit [33], Qspace [34], peps-torch [35], ad-peps [36], variPEPS [37], PEPSKit [38], TenNetLib [39].

In the following sections, we outline the design principles of YASTN and present a set of benchmarks demonstrating the computational speed-up from abelian symmetries. We focus on variational optimization of iPEPS for SU(2)-symmetric spin-$\frac{1}{2}$ model, SU(3)-symmetric model, and observables of a Hubbard model at finite temperature simulated via imaginary-time evolution.

## 2   Design principles

In this section, we give an overview of the structure of YASTN, presented in Fig. 2, and comment on some aspects of implementation. The basic building block of the library is the `yastn.Tensor` which is defined by the symmetry structure and the backend. The symmetry structure determines a set of allowed blocks and how to manipulate them when performing tensor algebra. The backend handles the execution of dense linear algebra operations and storage of tensor elements. These two are independent of each other. Symmetric tensors are used to construct TN ansätze, such as MPS and PEPS, and to finally define high-level algorithms that are applied to specific TN. For a detailed description of the library and all its functionalities, see the documentation under Ref. [22].

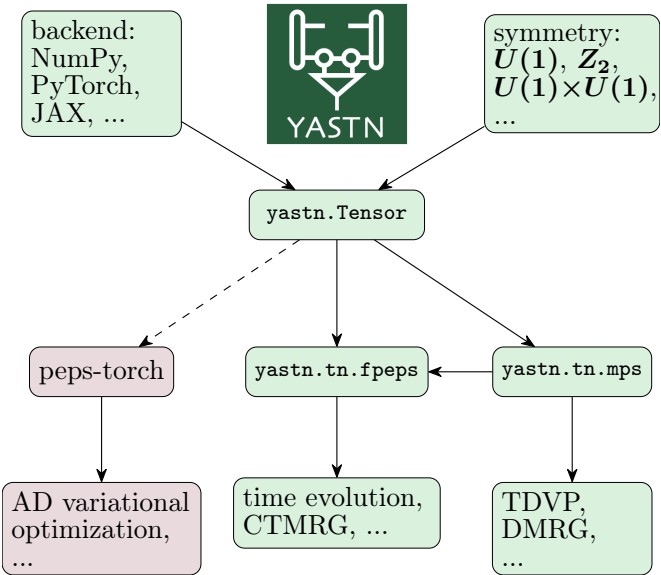

Figure 2: **Schematic design of** `yastn` **package.** The core element of the package is `yastn.Tensor`, which implements block-sparse tensor structure corresponding to a given abelian symmetry on top of a dense linear algebra backend, such as NumPy [40] or PyTorch [41]. More complex tensor networks, built on top of symmetric tensor, include standard but versatile MPS toolbox and 2D PEPS implementations to simulate the time evolution or ground-state variational optimization. The latter can benefit from automatic differentiation supported by some underlying backends, such as PyTorch.

## 2.1 Abelian-symmetric tensor

Tensors are multilinear maps from products of several vector spaces

$$T: \quad V^1 \otimes V^2 \otimes V^3 \otimes \dots \to \mathbb{C}, \tag{1}$$

where $V^i$ is a vector space and $\otimes$ is a tensor product. In a quantum-mechanical context, we work with Hilbert spaces $\mathcal{H}$ and their duals $\mathcal{H}^*$. By choosing some bases in each of these spaces the tensors can be written out in components

$$T = \sum_{abc\dots ijk\dots} T^{abc\dots}_{ijk\dots} |i\rangle |j\rangle |k\rangle \dots \langle a| \langle b| \langle c| \dots, \tag{2}$$

where $i, j, k \dots$ are indices of bases in $\mathcal{H}$ spaces, $a, b, c \dots$ in $\mathcal{H}^*$ spaces, and $T^{abc\dots}_{ijk\dots}$ is the corresponding tensor element. The action of the element $g$ of abelian group $G$ on tensor $T$ can be represented in a proper basis by diagonal matrices $U^i(g)$ transforming the tensor elements

$$
\begin{aligned}
(g \circ T)^{ab\dots}_{ij\dots} = \sum_{a'b'\dots i'j'\dots} & T^{a'b'\dots}_{i'j'\dots} [U^1(g)]^{i'}_i [U^2(g)]^{j'}_j \dots \\
& \times [U^m(g)^*]^a_{a'} [U^{m+1}(g)^*]^b_{b'} \dots.
\end{aligned}
\tag{3}
$$

The matrix elements of $U^i(g)$ are

$$[U^i(g)]^j_{j'} = \delta_{jj'} \exp(-i\theta_g t^{[i]}_j), \tag{4}$$

forming a diagonal matrix of complex phases defined by integer charges $t_j^{[i]}$, with angle $\theta_g \in [0, 2\pi)$ which depends on $g \in G$, and $\delta_{jj'}$ being Kronecker delta. Therefore, under the action of $g \in G$, each tensor element simply acquires a phase given by the sum of charges

$$(g \circ T)_{ij\ldots}^{ab\ldots} = T_{ij\ldots}^{ab\ldots} \exp[-i\theta_g(t_i^{[1]} + t_j^{[2]} + \ldots \\ - t_a^{[m]} - t_b^{[m+1]} - \ldots)]. \tag{5}$$

This form of the transformation gives a simple selection rule, a charge conservation, on the elements of symmetric tensors

$$t_i^{[1]} + t_j^{[2]} + \ldots - t_a^{[m]} - t_b^{[m+1]} - \ldots = n. \tag{6}$$

The charge of each non-zero element $T_{ij\ldots}^{ab\ldots}$ of a symmetric tensor must be $n$. In the case of $n = 0$, the tensor is invariant (unchanged) under the action of the symmetry. Otherwise, it transforms covariantly as all its elements are altered by the same complex phase $\exp(-i\theta_g n)$. The charges $t_j^{[i]}$ and $n$ and the precise form of their addition depend on the abelian group. For elementary abelian groups such as the cyclic group $C_N$ (often denoted $\mathbb{Z}_N$) the individual charges $t_j^{[i]}$ are elements of $\mathbb{Z}_N$, since in this case the allowed angles $\theta_g$ are integer multiples of $2\pi/N$, while for the circle group $U(1)$ the charges are elements of $\mathbb{Z}$. For direct products of abelian groups, charges become tuples $\vec{t}_j^{[i]}$ in the corresponding product of $\mathbb{Z}_N$'s and $\mathbb{Z}$'s.

By ordering the basis elements in each Hilbert space by their charge, the tensor $T_{ij\ldots}^{ab\ldots}$ naturally attains a block-sparse structure, which is central to the computational advantage offered by abelian-symmetric tensor network algorithms.

At the core of YASTN is the implementation of the symmetric tensor `yastn.Tensor`, as outlined in Refs. [17–19]. It is defined jointly by symmetry (block) structure data and tensor elements of existing blocks. First, we define a vector space with a charge structure, a `yastn. Leg`, determined by a signature $s = \pm1$ (distinguishing between $\mathcal{H}$ and dual $\mathcal{H}^*$), its charge sectors $\vec{t}$, and their corresponding dimensions $\vec{D}$, now decorated with arrows to denote tuples,

$$V(s, \vec{t}, \vec{D}) = \oplus_{\rho \in \vec{t}} \mathbb{C}^{\vec{D}_\rho}, \tag{7}$$

where $\rho$ now enumerates different charges instead of basis [1]. This space is a direct sum of simple spaces $\mathbb{C}^{\vec{D}_\rho}$, dubbed charge sectors. The effective dimension of such space is the sum of dimensions of individual charge sectors

$$D = \sum_\rho \vec{D}_\rho. \tag{8}$$

In the remainder of the text, we will refer to $D$ as the bond dimension when discussing scaling of computational complexity or memory requirements of TN algorithms with symmetric tensors. The abelian symmetric tensor of rank-$N$ is specified by the product of $N$ such vector spaces

$$T : \otimes_{i=1}^N V(s^{[i]}, \vec{t}^{[i]}, \vec{D}^{[i]}) \to \mathbb{C}. \tag{9}$$

The following is an example creating a random $U(1)$-symmetric tensor with total charge $n = 0$ with specified legs [2]:

---

[1]The conjugation of the leg, i.e., mapping space $\mathcal{H}$ to its dual space $\mathcal{H}^*$, is equivalent to flip of the signature and complex conjugation of elements.

[2]One can always define tensors with an extra dummy leg $V(-1, (n,), (1,))$, having a single charge sector of a unit dimension, making it invariant under symmetry transformations.

```
125 1 import yastn
126 2 from yastn.backend import backend_np
127 3 from yastn.sym import sym_U1
128 4
129 5 u1 = yastn.make_config(sym=sym_U1, backend=backend_np)
130 6 l = yastn.Leg(u1, s=1, t=(-1,1), D=(1,1))
131 7 lc = l.conj()
132 8 H = yastn.rand(u1, legs=[l,l,lc,lc], n=0)
```

133 which is, for example, compatible with a Hamiltonian $H = \vec{S}_1 \cdot \vec{S}_2$ of two spin-$\frac{1}{2}$ degrees of
134 freedom. The configuration created by `yastn.make_config` specifies the symmetry, e.g., `yastn`
135 `.sym.sym_U1` for the $U(1)$ in the example, and dense linear algebra backend (see below), e.g.,
136 `yastn.backend.backend_np` for NumPy.

137 The covariant transformation property of $T$ is imposed by the charge conservation of non-
138 zero blocks. Any block of tensor $T$ can be identified by selecting a charge sector $\rho_i \in \vec{t}^{[i]}$ on
139 each of the legs, i.e., a $N$-tuple of charges $(\rho_0, \rho_1, ..., \rho_N)$. All non-zero blocks must satisfy

$$\sum_{i=1}^{N} s^{[i]} \rho_i = n, \tag{10}$$

140 which is the block-sparse version of the element-wise charge conservation rule of Eq. (6).
141 Finally, we remark on the storage of tensor elements. In YASTN, the block data is initialized
142 *lazily*. The storage is allocated only for the blocks that have been assigned a non-zero value,
143 i.e., blocks allowed by the charge conservation but not assigned any value are not stored. All
144 allocated blocks are serialized together in a 1D array of dense linear algebra backend.

## 145 2.2 Fusion and contractions

146 The key operations on symmetric tensors, necessary for manipulating tensor networks, are
147 tensor reshape and permutation, commonly dubbed *fusion* in this context, and tensor contrac-
148 tions. Fusion resolves the tensor product of several spaces as a new space, i.e., the fusion of
149 legs into a new leg. Unlike reshaping the dense tensor, the shape cannot be freely chosen.
150 Instead, it is determined by the structure of the fused spaces. In particular, fusion orders and
151 accumulates tensor products of charge sectors on selected legs into new charge sectors under
152 the joint leg

$$V(s^{[i]}, \vec{t}^{[i]}, \vec{D}^{[i]}) \otimes V(s^{[j]}, \vec{t}^{[j]}, \vec{D}^{[j]}) \rightarrow V(s^{[r]}, \vec{t}^{[r]}, \vec{D}^{[r]}), \tag{11}$$

153 with new charge sectors $\vec{t}^{[r]}$ given by the unique combinations of charges $\vec{t}^{[i]} \otimes \vec{t}^{[j]}$

$$\vec{t}^{[r]} := \{v = s^{[r]}(s^{[i]}\rho + s^{[j]}\rho') : \rho \in \vec{t}^{[i]}, \rho' \in \vec{t}^{[j]}\}. \tag{12}$$

154 The dimension of new charge sector $v \in \vec{t}^{[r]}$ is [3]

$$\vec{D}_v^{[r]} = \sum_{\substack{\rho, \rho' \\ v = s^{[r]}(s^{[i]}\rho + s^{[j]}\rho')}} \vec{D}_\rho^{[i]} \vec{D}_{\rho'}^{[j]}. \tag{13}$$

155 The fusion and un-fusion calls are demonstrated below on previously constructed rank-4 tensor
156 $H$, first fusing pairs of legs resulting in a matrix form

```
157 1 H_mat = H.fuse_legs(axes=((0,1), (2,3)))
158 2 H = H_mat.unfuse_legs(axes=(0,1))
```

---

[3]Following a *lazy* approach adopted in YASTN, a new fused leg contains only charges for which some non-zero
tensor block exists. As such, fusion in YASTN is always done in the context of particular tensor.

159  where the `axes` argument of `fuse_legs` is a tuple that describes how the legs should be
160  permuted and fused in the resulting tensor. Following NumPy convention, each leg of the
161  original tensor is labeled by an integer according to its position starting at 0. The groups of
162  legs to be fused are specified by nested tuples in `axes`. Several alternative ways of fusing $H$
163  are given below

```
1 H_long_mat = H.fuse_legs(axes=(0, (1,2,3)))
2 H_thin_mat = H.fuse_legs(axes=((0,1,2), 3))
3 H_thin_mat_transposed = H.fuse_legs(axes=(3, (0,1,2)))
4 H_rank3_permuted = H.fuse_legs(axes=(0, (3,2), 1))
```

168  When fusing legs, YASTN first calculates the structure of the resulting tensor with fused leg(s),
169  i.e., the tuple $(s^{[r]}, \vec{t}^{[r]}, \vec{D}^{[r]})$ and a set of dense linear algebra jobs (permutes, reshapes, and
170  copies) to be executed by the backend to populate the new 1D storage array with serialized
171  blocks. The resulting tensor records the original structure, and hence, the fusion can be re-
172  verted by `unfuse_legs`.

```
1 H_1fusionlevel = H.fuse_legs(axes=(0,(1,2),3))
2 H_2fusionlevels = H_1fusionlevel.fuse_legs(axes=(0,(1,2))
3 H = H_2fusionlevels.unfuse_legs(axes=1).unfuse_legs(axes=1)
```

176  The tensor contraction of symmetric tensors is realized by a commonly adopted workflow.
177  First, the tensors are fused into block-sparse matrices [4], then matrix-multiplied along the con-
178  tracted legs, and finally unfused to obtain the desired form:

$$\sum_{x_0 x_1 \dots} A_{i_0 i_1 x_0 i_2 x_1 i_3 x_2 \dots} B_{x_1 j_0 x_0 x_2 j_1 \dots} \xrightarrow{\text{permute}} \sum_{\{x\}} A_{\{i\} \cup \{x\}} B_{\{x\} \cup \{j\}} \xrightarrow{\text{reshape}} \sum_X A_{IX} B_{XJ} \xqquad\overset{\text{multiply}}{=}$$

$$= C_{IJ} \xrightarrow{\text{unfuse}} C_{\{i\}\{j\}} = C_{i_0 i_1 i_2 \dots j_0 j_1 j_2 \dots} \tag{14}$$

179  where $\{x\}$ is a set of common legs that become fused into single leg $X$, and original uncon-
180  tracted legs $\{i\}$ and $\{j\}$ are restored from the fused $I$ and $J$ to obtain the final tensor. We
181  again adhere to NumPy convention, where `axes` argument of `tensordot` contains two tuples.
182  The first tuple specifies legs to contract on the first tensor operand $A$, and the second tuple
183  specifies legs to contract on the second tensor operand $B$. The legs in the same position in
184  the two tuples are then contracted, i.e. `axes=((2,4,6,...),(2,0,3,...))` for the above
185  example. The relative order of the remaining uncontracted legs of the first operand $\{i\}$ and
186  the second operand $\{j\}$ is unchanged. The legs of the resulting tensor $C$ are ordered exactly
187  as the uncontracted legs of the two tensor operands. First, we have the uncontracted legs $\{i\}$
188  of the first operand followed by the uncontracted legs of the second operand $\{j\}$. Alterna-
189  tive contraction workflows, which avoid the fusion to the block-sparse matrix form, will be
190  introduced in the future version.

191  Here, we first show an example call for pairwise tensor contraction, and second, an equiv-
192  alent given in terms of explicit operations

```
1 H2 = yastn.tensordot(H, H, axes=((2,3), (0,1)))
2 H2 = (H.fuse_legs(axes=((0,1), (2,3))) @
3        H.fuse_legs(axes=((0,1), (2,3)))).unfuse_legs(axes=(0,1))
```

196  For both fusion and multiplication, the YASTN first precomputes what the non-zero blocks are,
197  so the backend performs only the relevant operations. Contractions of more general tensor
198  diagrams are supported through convenience functions, such as `einsum` and `ncon` [42] (in our
199  case differing only by syntax), which are based on the elementary operations discussed above.

---

[4]For valid contraction, the structures of the contracted legs must be compatible, including the origins of any
fused leg. YASTN automatically resolves a situation when some charges in the fusion history of a to-be-contracted
fused leg are missing but are present in its contraction partner. This is done by utilizing information on the tensor's
fusion history stored in each `yastn.Tensor` object.

## 2.3 Tensor network algorithms

The symmetric tensor serves as the basis for higher-level tensor network structures and algorithms. Here, YASTN comes with MPS and PEPS modules. The MPS module supports finite-size MPS with the implementation of a range of standard algorithms, including DMRG for ground-state optimization, TDVP [43,44] for time evolution [5], and overlap maximization [46] against a general target, i.e., MPS, sum of MPSs, or sum of MPO-MPS products. This is complemented by a versatile high-level (Hamiltonian) MPO generator. The MPS module provides subroutines for some PEPS methods; e.g., it was utilized in Ref. [47] for boundary MPS contraction and long-range correlations calculation in a finite PEPS defined on a cylinder. At the same time, it is a versatile computational toolbox on its own. For example, it has been used in simulations of Lindbladian dynamics in the context of fermionic quantum transport [48], where the $U(1)$ symmetry reflects a lack of correlations between different particle-number sectors of a density matrix.

The PEPS module features the implementation of fermionic PEPS, dubbed fPEPS (which also allows simulations of systems without fermionic statistics). It covers both finite PEPS defined on a square lattice and its infinite versions for translationally invariant (over a unit cell) systems in the thermodynamic limit. It supports a range of time evolution algorithms, starting with *neighborhood tensor update* (NTU) scheme [49,50], its refinement to a family of larger environmental clusters [47], ending on a full-update type of schemes [16,51]. It is viable for imaginary-time evolution, e.g., in the context of finite-temperature simulation of density matrix purification [52] or minimally-entangled typical thermal states [53], and real-time simulations, e.g., of pure-state quench-dynamics in disordered spin systems [47].

# 3 Examples

To demonstrate the use and versatility of YASTN, we present three end-to-end numerical examples[6] centered on iPEPS. We provide benchmarks of MPS-type contractions and algorithms with comparisons to ITensor and TeNPy in a dedicated repository [54].

We show computational speed-up and reduced memory footprint obtained with YASTN by utilizing abelian symmetries for the following examples:

1. Sec. 3.1: variational optimization of iPEPS with $D \leq 8$ for antiferromagnetic spin-$\frac{1}{2}$ model on a square lattice using $U(1)$ symmetry,

2. Sec. 3.2: variational optimization of iPEPS with $D \leq 13$ for SU(3) model on Kagome lattice using $U(1) \times U(1)$ symmetry,

3. Sec. 3.3: observables of thermal iPEPS of Hubbard model at finite temperature using $\mathbb{Z}_2$, $U(1)$, and $U(1) \times U(1)$ symmetry, with $D$ up to $36$ for the latter.

In Sec. 3.1 and Sec. 3.2 we optimize iPEPS for SU(2) model on square and SU(3) model on Kagome lattices, respectively. First, given an iPEPS generated by a set of tensors $\vec{a} = \{a, b, \ldots\}$, we compute an approximate environment tensors $\vec{E}(\vec{a})$ (specified below) with the precision governed by the environment dimension $\chi$. Then, the environment $\vec{E}$ and the tensors $\vec{a}$ are combined to evaluate the energy per site $e$ of the Hamiltonian. Finally, the reverse mode of AD (i.e., backpropagation) is invoked to calculate the gradient $\partial e / \partial \vec{a}$. The most computationally intensive stage is the construction of environments, scaling as the cube of $D^2 \chi$, which assuming

---

[5]In TDVP, we employ adaptive Krylov subspace exponential integrator of Ref. [45]

[6]Examples 1 and 2 were run on a single Intel® Xeon® Silver 4110 processor. Example 3 was run on a single Intel® Xeon® Gold 6416H processor

241 the necessary $\chi \propto D^2$ gives the overall complexity $\mathcal{O}(D^{12})$, where $D$ is the bond dimension of
242 the iPEPS tensors. We use iPEPS optimization implemented in peps-torch [35], here operating
243 on the YASTN symmetric tensors.

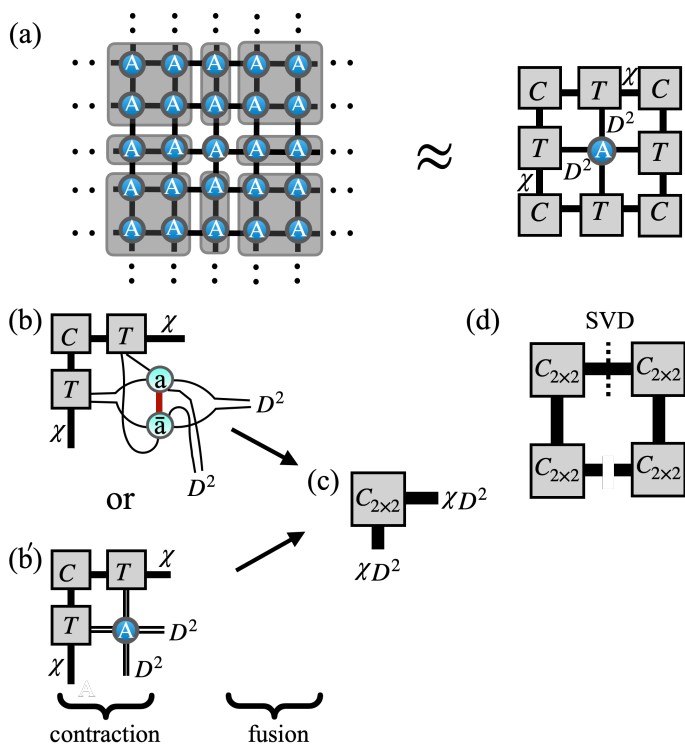

Figure 3: **Corner transfer matrix iteration**. In (a), CTM approximates parts of
an infinite tensor network by a set of finite environment tensors characterized by
environment bond dimension $\chi$. We depict elements of the CTM algorithm step (it-
eration) that dominate the computational effort shown in Figs. 4–6. Panels (b), (b'),
and (c) depict the construction of an enlarged corner combining CTM environment
tensors (rectangular) with PEPS tensors (circle). Panel (d) shows an SVD decompo-
sition of a product of four enlarged corners that is then used to construct the CTM
projections from enlarged virtual spaces.

244 For the examples presented, we employ the *corner transfer matrix* (CTM) algorithm [55–
245 57] to compute the environments. CTM approximates environments $\vec{E} = \{C, T\}$ of iPEPS by a
246 set of $\chi \times \chi$ corner matrices $C$ and $\chi \times D^2 \times \chi$ transfer tensors $T$, as shown in Fig. 3(a). Alter-
247 natively, one can use the boundary MPS methods [16, 46, 58]. The computational complexity
248 of CTM arises from two sources (see Fig. 3), tensor contractions, and *singular value decompo-*
249 *sition* (SVD) when computing low-rank approximations, both scaling as $\mathcal{O}(D^{12})$. In practice,
250 for simulations without symmetries, the SVD gives a substantially greater contribution due to
251 a higher scaling prefactor and a poor speed-up offered by multithreading or GPU acceleration
252 compared to tensor contractions. However, for symmetric iPEPS the situation becomes more
253 nuanced, as we demonstrate in the following examples.

254 In Sec. 3.3, the purification techniques are applied to compute thermal expectation values
255 for the Fermi-Hubbard model on a 2D square lattice. To effectively transform the thermal
256 density matrix into a purified wave function, we use the ancilla trick and perform imaginary
257 time evolution to reach the target temperature. We adopt the NTU algorithm to optimize the
258 time evolution, with computational cost scaling of $\mathcal{O}(D^8)$ dominated by tensor contractions.
259 We focus on the final calculation of the expectation values using CTM with $\chi = 5D$, translating
260 to $\mathcal{O}(D^9)$ scaling. We quantify the sources of advantage offered by incorporating various

261  symmetries.

## 3.1 Heisenberg antiferromagnet with anisotropy

263  We revisit the Heisenberg model with anisotropy in the couplings describing a system of cou-
264  pled spin-$\frac{1}{2}$ ladders

$$\mathcal{H} = J \sum_R \mathbf{S}_R \cdot \mathbf{S}_{R+\hat{x}} + \sum_R J_R \mathbf{S}_R \cdot \mathbf{S}_{R+\hat{y}}, \tag{15}$$

265  where $\mathbf{S}_R = (S_R^x, S_R^y, S_R^z)$ is the $S = \frac{1}{2}$ operator at site $R = (x, y)$ on a square lattice spanned by
266  the $\hat{x}$ and $\hat{y}$ unit vectors. The coupling $J_R = J$ for the odd position and $J_R = \alpha J$ for the even
267  position along the $y$ axis, respectively. For $\alpha = 1$, the Hamiltonian in Eq. (15) realizes the
268  Heisenberg model on the square lattice, and for $\alpha = 0$ it corresponds to a system of decoupled
269  two-legged ladders. The model was previously addressed with iPEPS in Ref. [59]. We adopt
270  the same description that uses 2×2 unit cell with four non-equivalent tensors $\vec{a} = \{a, b, c, d\}$ [7].

271      The ground states possess $U(1)$ symmetry corresponding to the conservation of the $S^z$
272  component. The symmetry can be exploited by utilizing $U(1)$-symmetric iPEPS. The results,
273  summarized in Fig. 4, show a rapidly growing computational advantage of symmetric iPEPS
274  for bond dimensions $D > 4$. While at $D = 4$ the overhead due to block-sparse logic is still
275  significant, at the largest bond dimension considered, $D = 8$, a 30-fold speed-up is observed.

276      In practical terms, the convergence of the CTM towards the desired precision, here mea-
277  sured by the error on the energy per site becoming lower than $\epsilon < 10^{-8}$, typically requires
278  $\mathcal{O}(10)$ iterations; thus without $U(1)$ symmetry a single optimization step would already take
279  hours. Details of the block-sparse structure and its impact on the CTM are visualized in
280  Fig. 4(b,c). At the largest bond dimension, $D = 8$, the fusion of the enlarged corner into
281  a block-diagonal matrix requires processing of approximately $\mathcal{O}(100)$ blocks by performing
282  dense permutes, reshapes, and copies, with the largest block having $\mathcal{O}(10^5)$ elements. The
283  cost of subsequent SVD is dominated by the largest block(s) of fused enlarged corner, which
284  are $L \times L$ matrices with $L = 1,500 \sim 2,000$. As a result, computational time contributions are
285  shared between SVD and tensor contractions with fusions roughly as 3:2, with SVD being the
286  dominant factor.

## 3.2 SU(3) model on Kagome lattice

288  We consider an SU(3)-symmetric model on Kagome lattice, analyzed recently in Ref. [60],
289  where each site holds a single degree of freedom from the fundamental representation **3** of
290  the SU(3) group spanned by three states $\{|\alpha\rangle, |\beta\rangle, |\gamma\rangle\}$. The Hamiltonian reads

$$H = J \sum_{\langle i,j \rangle} P_{ij} + \sum_{\triangle ijk} (K P_{ijk} + h.c.), \tag{16}$$

291  where $P_{ij}$ is a permutation, $P_{ij}|\alpha\rangle_i|\beta\rangle_j = |\beta\rangle_i|\alpha\rangle_j$, of local states on nearest-neighbour
292  bonds. $P_{ijk}$ is a clockwise permutation of local states in nearest-neighbor triangles such that
293  $P_{ijk}|\alpha\rangle_i|\beta\rangle_j|\gamma\rangle_k = |\gamma\rangle_i|\alpha\rangle_j|\beta\rangle_k$, with fixed choice of orientation of triangles, and $J$ and $K$ are
294  real and complex-valued couplings, respectively.

295      In this section, we demonstrate an advantage of $U(1) \times U(1)$-symmetric iPEPS, utilizing
296  maximal abelian subgroup of SU(3), over implementation without symmetries [8]. To compute

---

[7]A more efficient description might generate all tensors in 2×2 unit cell from a single parent tensor $a$ by use of
unitary $-i\sigma^y$ acting on physical index and/or permutation of virtual indices generated by the reflection along the
$x$-axis. However, such a parameterization would not change the complexity of the CTM

[8]In this example, besides iPEPS, one can also use different ways to construct two-dimensional TN ansatz on
Kagome lattice, i.e., infinite projected simplex states (iPESS), however, the computational complexity $\mathcal{O}(D^{12})$,
attributable to CTM, remains unchanged.

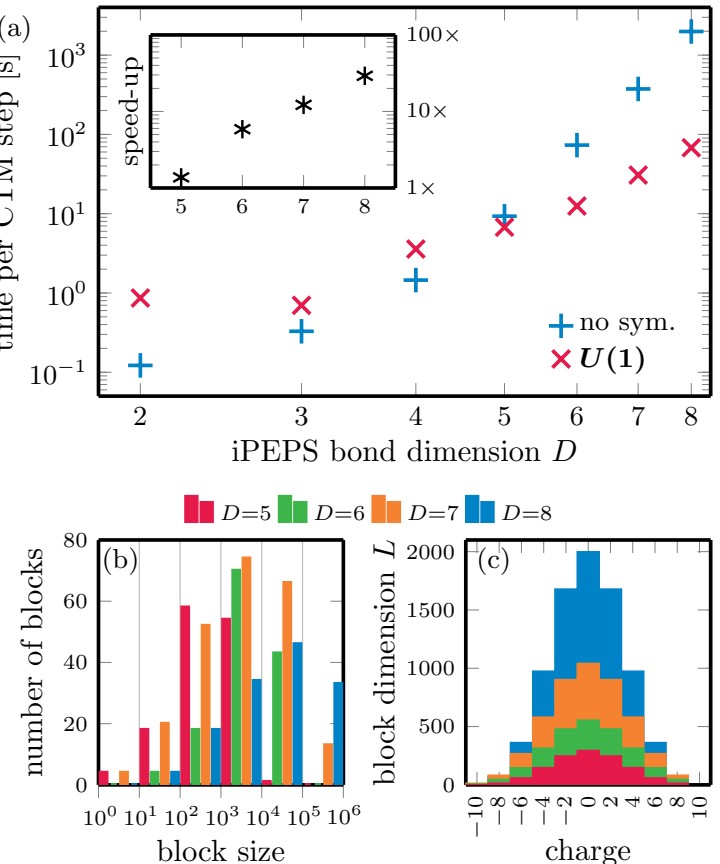

Figure 4: **Use case 1: optimization of $U(1)$-symmetric iPEPS for model of spin-$\frac{1}{2}$ coupled ladders.** In (a), we show the scaling of the wall time per CTM step (in seconds) for the entire gradient optimization step of iPEPS. The bond dimension of the environment is $\chi = 2D^2$. The inset shows the relative speed-up compared to an implementation without symmetry. In (b), we show a distribution of blocks of an enlarged corner by their size (number of elements) before fusion to a block-diagonal matrix as shown in Fig. 3(c). In (c), we show the sizes of $L \times L$ blocks after the fusion.

CTM environments on the Kagome lattice, we coarse-grain three sites on each down-pointing triangle into a single tensor, resulting in an effective square lattice. The local Hilbert space dimension thus grows to $3^3 = 27$, making the optimizations memory intensive. In Fig. 5, we demonstrate the dramatic speed-up achieved by utilizing $U(1) \times U(1)$ symmetry. For $D = 9$ iPEPS, a single gradient step is already accelerated by more than a factor of **100**. For larger bond dimensions, the simulations without symmetries become prohibitive, and we estimate the speed-up based on extrapolation of the scaling at smaller bond dimensions.

In contrast to the example in Sec. 3.1 utilizing the $U(1)$-symmetry, the speed-up in this case is not monotonic in $D$. This happens because of the varying structure of the iPEPS tensors, i.e., the allowed symmetry sectors and their sizes. In Fig. 5(b,c) we illustrate the block structure of enlarged corners before and after the fusion to a block-diagonal matrix. Generally, for larger groups, the number of blocks of enlarged corners before fusion is substantially higher. Even at $D = 7$, the total number of blocks is already more than **3,000** whereas for $U(1)$-symmetric enlarged corner in Sec. 3.1 it was below **300**. For $D = 13$ ansatz, the fusion of the enlarged corner into a block-diagonal matrix requires processing of more than **16,000** blocks, with more than half of them being small in size, having roughly $\mathcal{O}(10^3)$ elements or less. This granularity defines the bottleneck of the simulations. For $D = 12$ and $D = 13$ the ratios

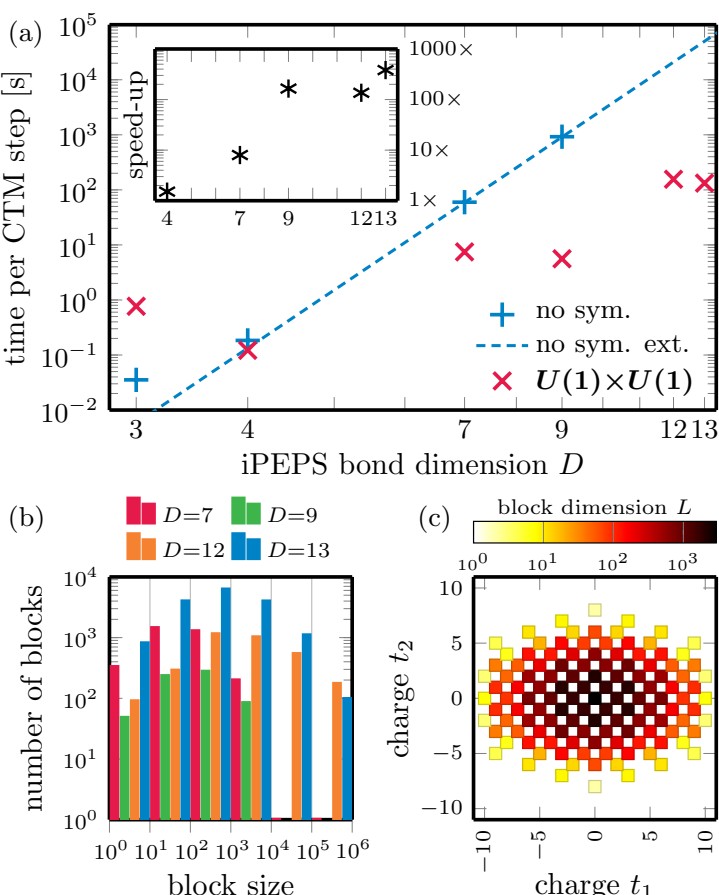

Figure 5: **Use case 2: optimization of $U(1)\times U(1)$-symmetric iPEPS for SU(3) Kagome model.** In (a), scaling of the wall time per CTM step (in seconds) for the entire gradient optimization step of iPEPS. The bond dimension of the environment is $\chi = D^2$. The inset shows the relative speed-up compared to an implementation without symmetry, with $D = 12$ and $13$ simulation wall times estimated from the extrapolation (blue dashed line). In (b), a distribution of blocks of an enlarged corner by their size (number of elements) before the fusion to a matrix. In (c), $L \times L$ blocks of the block-diagonal enlarged corner after fusion. We plot them as a heatmap, with different $U(1)$ charges on x- and y-axes.

between the computational time of SVD and contractions including fusion to block-diagonal enlarged corners are 3:4 and 1:10, respectively. Overall, the $U(1)\times U(1)$ simulations become dominated by fusion, with SVD being sub-leading. The precise speed-up depends on the sizes of the blocks, such as here, where $D = 12$ has a slightly higher proportion of largest blocks compared to $D = 13$.

### 3.3 2D Fermi-Hubbard model on a square lattice

We consider a two-dimensional Fermi-Hubbard model (FHM) with on-site repulsion as studied in Ref. [52]. The Hamiltonian reads

$$H = -t \sum_{\langle i,j\rangle,\sigma} \left( c_{i\sigma}^{\dagger} c_{j\sigma} + c_{j\sigma}^{\dagger} c_{i\sigma} \right) + U \sum_{i} \left( n_{i\uparrow} - \frac{1}{2} \right)\left( n_{i\downarrow} - \frac{1}{2} \right) - \mu \sum_{i} n_i, \qquad (17)$$

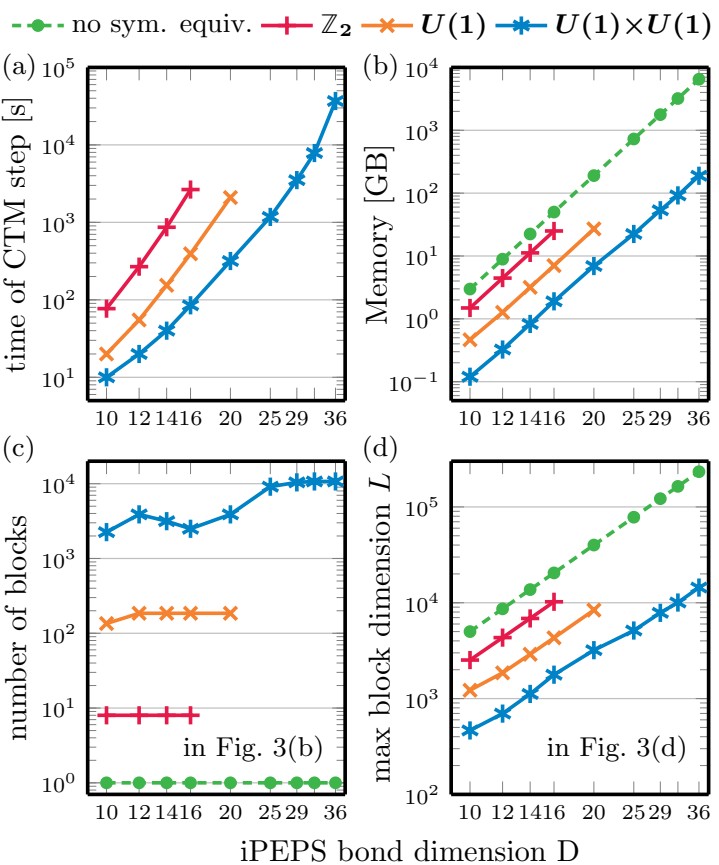

Figure 6: **Use case 3: expectation values from CTM environments in fermionic iPEPS for finite-temperature Hubbard model.** The bond dimension of the environment is $\chi = 5D$, which is sufficient here to converge the expectation values. We show, in (a), the wall time per CTM step and, in (b), the data size (memory requirement) of the biggest intermediate tensor appearing while building enlarged CTM corners in Fig. 3(b). Panel (c) shows the number of blocks in an enlarged corner before the fusion in Fig. 3(c), and panel (d) is the size of the largest $L \times L$ block for SVD in Fig. 3(d).

where $c_{i\sigma}^{\dagger}$ and $c_{i\sigma}$ are the creation and annihilation operators for an electron with spin $\sigma$ at site $i$, $n_{i\sigma} = c_{i\sigma}^{\dagger} c_{i\sigma}$ is a corresponding number operator, $t$ is the hopping amplitude, $U$ is the on-site Coulomb repulsion strength, and $\mu$ is the chemical potential.

The iPEPS ansatz employs a checkerboard lattice with a 2-site unit cell. The thermal state for the inverse temperature $\beta$ is obtained by evolving the infinite-temperature purification $|\psi(0)\rangle$ under the non-unitary propagator $U(\beta) = e^{-\beta H/2}$. The initial purification is a product of maximally entangled pairs between each physical site and its corresponding ancilla, $|\psi(0)\rangle = \prod_j \prod_{m=\uparrow,\downarrow} \frac{1}{\sqrt{2}} \sum_{s_{m_j}=a_{m_j}=0,1} |s_{m_j} a_{m_j}\rangle$, translating to local Hilbert spaces of dimension $4^2 = 16$. In the following examples, we run the imaginary time evolution employing the NTU scheme targeting $\beta = 2$.

In YASTN, the fermionic exchange order is implemented following the scheme of Refs. [61, 62] by projecting the lattice ansatz onto a plane, imposing a canonical fermionic order, and applying swap gates on every line crossing; see Fig. 3(b). The swap gate introduces sign changes for blocks with charges of odd parity on both swapped legs. This makes $\mathbb{Z}_2$ the minimal symmetry needed for fermionic system simulations. The Hamiltonian in Eq. (17) preserves

the number of particles per spin direction, which allows us to implement the model under $U(1) \times U(1)$ symmetry as the highest abelian symmetry.

The expectation values of the thermal state at $\beta = 2$ are calculated using the CTM. Fig. 6 shows an advantage of symmetric tensors by comparing $\mathbb{Z}_2$ (parity; minimal requirement for fermionic statistics), $U(1)$ (total charge conservation) and $U(1) \times U(1)$ (total charge conservation for each spin) symmetries. We also show equivalent values for the corresponding tensors with no symmetry. We choose the environmental bond dimension $\chi = 5D$, which is sufficient to converge the expectation values in this example.

Fig. 6(a) presents the computational wall time for a CTM step as a function of the iPEPS bond dimension for all the symmetries tested, with the systematic improvement offered by higher symmetries. Fig. 6(b) highlights the memory usage bottleneck, showcasing the size of the largest object formed during the CTM iteration, i.e., an intermediate step of the contraction in Fig. 3(b); this tensor has to be later fused, and there are other tensors in the memory, so the memory peak is roughly two times higher. This illustrates that those simulations are ultimately memory-limited. Employing $U(1) \times U(1)$ symmetry offers a systematic **30**-fold memory gain compared to tensors without symmetries, which ultimately allows for successful simulations up to $D = 36$.

Following the previous examples, in Fig. 6(c), we present the number of blocks processed during the fusion that form enlarged corners in Fig. 3(c). A particular challenge for $U(1) \times U(1)$ case is the number of blocks that can exceed **10, 000**. However, SVD is a dominant factor that takes at least half the simulation time for $D \geq 25$ in our numerical experiments. In Fig. 6(d), we show the (sectorial) bond dimension of the largest block decomposed in Fig. 3(d), which is $\mathcal{O}(10^4)$ for each employed symmetry. Among them, the $U(1) \times U(1)$-symmetry offers here a **15**-fold improvement for given $D$ as compared to a setup with no symmetries involved.

# 4 Conclusion and future outlook

Tensor networks are becoming increasingly popular tools for numerical treatment of quantum systems, ranging from ground-state simulations of condensed-matter systems to simulations of quantum circuits. The landscape of associated software is continuously growing. For 1D and quasi-1D geometries, well-established and mature packages offer a rich set of MPS algorithms that cover direct energy minimization, (imaginary) time evolution, and much more. For two-dimensional geometries, predominantly targeted by iPEPS, the field remains nascent.

Here, we have introduced YASTN, a Python-based TN library with a strong emphasis on simulations of two-dimensional systems by iPEPS, which is motivated by the need for both abelian symmetries and automatic differentiation. By design, the dense linear algebra and the AD engine are provided by different backends, allowing for implementation-specific optimizations. YASTN, with its rich set of examples covering ground-state simulations of various 2D spin lattice models (through peps-torch) and finite-temperature simulations of the 2D Hubbard model, thus joins similar efforts by VariPEPS [37], PEPSkit [38], ad-peps [36], and peps-torch [35] together lowering the barrier for entry.

The wide separation between the high-level description of iPEPS algorithms and their fast execution, optimized down to low-level dense linear algebra, especially for symmetric tensors, remains a challenge. Unlike MPS simulations, iPEPS contraction algorithms for computing environments and evaluation of observables involve a more diverse set of tensor contractions, varying in ranks and block sparsity patterns. Furthermore, flexible deployment and the ability to leverage heterogeneous clusters, which account for the iPEPS-specific block sparsity, is vital to address the sharp $\mathcal{O}(D^8 \sim D^{12})$ (albeit polynomial) scaling with the bond dimension, which is the key resource governing the precision of iPEPS. Thus, these challenges call for further

384 development.

# Acknowledgements

386 We thank Philippe Corboz, Piotr Czarnik, Jacek Dziarmaga, Boris Ponsioen, Yintai Zhang and
387 Yi Xu for inspiring discussions that were invaluable in the development of this package.

388 **Funding information**    We acknowledge the funding by the National Science Center (NCN),
389 Poland, under projects 2019/35/B/ST3/01028 (A.S.), 2020/38/E/ST3/00150 (G.W.), and
390 project 2021/03/Y/ST2/00184 within the QuantERA II Programme that has received funding
391 from the European Union Horizon 2020 research and innovation programme under Grant
392 Agreement No. 101017733 (M.M.R.). J.H. acknowledges support from the European Research
393 Council (ERC) under the European Union's Horizon 2020 research and innovation programme
394 (grant agreement No 101001604) and from the Swiss National Science Foundation through a
395 Consolidator Grant (iTQC, TMCG-2_213805).

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
