# Peer review of "YASTN: Yet another symmetric tensor networks; A Python library for abelian symmetric tensor network calculations"

_SciPost Physics Codebases, doi:SciPost Phys. Codebases 52 (2025) , SciPost Phys. Codebases 52-r1.2 (2025)_

## Round 1 · Referee Report · Johannes Hauschild (Referee 1) · 2024-8-22

Strengths

1) easy to install 2) clear code structure 3) extensive test suite 4) pytorch backend with support of GPU acceleration and auto differentation 5) interface quite similar to well-known numpy

Weaknesses

1) The benchmarks do not compare to other implementations, but only check the expected scaling within the package. It might be interesting to compare the overhead implied by charge conservation to other libraries like TeNPy or ITensor. If the authors are interested, I'd be happy to help setting up a comparison with TeNPy similar to https://github.com/tenpy/tenpy_benchmarks and https://itensor.github.io/ITensorBenchmarks.jl/dev/tenpy_itensor/index.html

2) The documentation includes installation instructions, examples and benchmarks as required by the acceptance criteria. However, it could be a bit more extensive in some places. E.g. https://yastn.github.io/yastn/examples/fpeps/ctmrg.html mentions a comparison to the Onsager solution. While this is indeed done in the corresponding tests/peps/test_ctmrg.py file, this comparison is not included in the docs.

Report

YASTN is a solid python library for tensor network calculations based on the widely adopted numpy and pytorch as backends. It extends those packages by implementing a tensor class exploiting a block-diagonal structure emerging from abelian symmetries, and the benchmarks in the manuscript demonstrate that using the symmetries can lead to significant speedups. The tensor interface is quite similiar to the well-known numpy package; in contrast to e.g. TeNPy it does not introduce labels, or make the legs globally unique like iTensor.
High-level tensor network algorithms are provided for (finite and infinite) PEPS (the focus of the library) and finite MPS.

While YASTN indeed joins a row of other tensor network libraries as implied by the name, it has sufficiently different focus than the other packages and is an independent implementation valuable for cross-checks. Further, the demand has already been demonstrated by several publications employing it for state-of-the-art calculations. I hence think that the acceptance criteria for Scipost Physcis Codebase are met and recommend a publication.

Requested changes

Optional: benchmark comparsion to other tensor network library, see point 1) under weaknesses.

Recommendation

Publish (meets expectations and criteria for this Journal)

  • validity: top
  • significance: good
  • originality: good
  • clarity: top
  • formatting: perfect
  • grammar: perfect

Author:  Juraj Hasik  on 2025-01-06  [id 5083]

(in reply to Report 1 by Johannes Hauschild on 2024-08-22)

We thank the referee for his review and we address the highlighted weaknesses below. We believe the revised manuscript and the updated YASTN repository together with a new repo dedicated to performance benchmarks improves upon these weak points.

1) The benchmarks do not compare to other implementations, but only check the expected scaling within the package. It might be interesting to compare the overhead implied by charge conservation to other libraries like TeNPy or ITensor. If the authors are interested, I'd be happy to help setting up a comparison with TeNPy similar to https://github.com/tenpy/tenpy_benchmarks and https://itensor.github.io/ITensorBenchmarks.jl/dev/tenpy_itensor/index.html

We have created a dedicated repository for performance benchmarks at https://github.com/yastn/yastn_benchmarks. It includes the suggested DMRG benchmark with comparisons to TenPy and ITensor, an iPEPS optimization benchmark featured in examples 1 and 2 of the manuscript, and a template for a representative set of symmetric tensor contractions discussed in this work. This repository serves as a point of reference for comparing current and future versions of the library.

2) The documentation includes installation instructions, examples and benchmarks as required by the acceptance criteria. However, it could be a bit more extensive in some places. E.g. https://yastn.github.io/yastn/examples/fpeps/ctmrg.html mentions a comparison to the Onsager solution. While this is indeed done in the corresponding tests/peps/test_ctmrg.py file, this comparison is not included in the docs.

We carefully iterated though the documentation to improve its clarity. Among many small changes in the documentation, the above-mentioned CTMRG example has been rewritten from scratch.

---

## Round 1 · Referee Report · Jutho Haegeman (Referee 2) · 2024-9-4

Strengths

1) Paper: interesting and instructive examples 2) Paper: good balance between conciseness and details 3) Software: one of the few packages with full support for GPUs and automatic differentiation 4) Software: one of the few public packages for simulations with (infinite and possibly fermionic) PEPS 5) Software: extensive set of examples and models (lattices and Hamiltonians)

Weaknesses

1) Some statements or code samples in the paper would benefit from a more extensive explanation (see report) 2) Somewhat unclear code structure regarding the interplay with peps-torch (see report) 3) No benchmark comparisons to 'competing' software packages 4) Grammar (use of articles) can be improved 5) Examples focus on PEPS (on CPUs) only

Report

This paper presents the open-source Python package YASTN for tensor network simulations of quantum many-body systems. It has built-in support for matrix product states (MPS) and, notably as compared to some of the other main packages in the field, extensive support for projected entangled-pair states (PEPS). At the core is a tensor type that can represent tensors with abelian symmetries and for which the data can be stored using both Numpy and PyTorch backends, where the latter enables support for GPUs and automatic differentation. These features, together with the extensive PEPS support, set it apart from its main competitor in the Python universe, namely TenPy (which arguably has more support for different MPS algorithms and extensions).

The manuscript itself reads well (aside from an occasional missing or ill-used article "a" / "the"). It concisely introduces the design principles, explaining in a good amount of detail how abelian symmetries constrain the number of nonzero tensor entries and how to specify such symmetries and create and manipulate tensors with them. I have a few small questions or clarification requets on this section which are listed below.

About the second half of the manuscript is spend on a number of impressive examples that are instructive regarding the implications of using abelian symmetries in PEPS simulations. However, given the extensive set of features of this package, the benchmarks do only cover a very small fraction of it, and in particular do not cover aspects such as * CPU versus GPU with and without symmetries, and the effect of the symmetry overhead in that case * comparison to other packages such as TenPy and ITensors.jl, for simple tensor network contraction or for ... next bullet point * anything related to MPS (possibly in relation to the previous bullet point, as the main competitors have a strong MPS emphasis) * memory and runtime cost of automatic differentiation

Requested changes

I will list a number of questions and suggestions that might result in changes here, but they are optional and could also be addressed in the response:

1) Around line104: it is perhaps worth mentioning that the charges for $Z_n$ are elements of $\mathbb{Z}_n$, because in those cases, the angle theta in the symmetry is restricted to take values that are integer multiples of $2\pi/N$. Also, I don't quite understand why the different symbol $Z_n$ and $\mathbb{Z}_n$ are used, but if the first one is supposed to refer to the multiplicative group, it is probably more correct to denote it as the cyclic group $C_n$.

2) Around line 113: it is stated that boldface is used to underline the vector nature, but it seems that the SciPost style makes all math in boldface, so that also scalar values such as $s$, individual dimensions per sector $D_\rho$, and esssentially all other math is typeset in boldface.

3) The final paragraph of section 2.1: can you elaborate on the the benefits of the lazy initalisation and the allocation of nonzero blocks only. Is it often the case that blocks which are allowed by the symmetry, turn out to be zero anyway? What happens if they receive nonzero data afterwards? Does this require to allocate a new tensor or to copy the data?

4) The code examples in lines 155-156 and 169-170 could perhaps use a bit more explanation, especially for non-Python users (although they might not be the typical audience of this paper). I was in particular wondering about what exactly the 'axes' argument in the 'tensordot' call specifies, and how the indices of the resulting tensor are sorted/defined. Another question is about the recording of the original structure of fusing legs: what happens if different fusion steps are concatenated, e.g. first 1 with 2 and 3 with 3, then the resulting 1 with the resulting 2. Are these intermediate steps recorded. Will 'unfuse_legs' take one step back, or immediately jump back to the original structure?

5) Given that wall times are reported in seconds, would it be possible to provide some details on the hardware on which these simulations were ran?

6) More benchmarks can provide further interesting insight, and as listed in the report section, there are quite a few benchmarks that could be conceivable. I leave it to the authors to decide what they believe could be useful additions here.

7) I believe it would be good if the interplay/interdependency with peps-torch could be clarified. It is sometimes listed as an independent package (introduction, line 67, as well as conclusion, line 348), but at the same time seems closely tied to YASTN and is mentioned as part of it for doing the PEPS optimisations (e.g. conclusion, line 347).

Recommendation

Publish (easily meets expectations and criteria for this Journal; among top 50%)

  • validity: top
  • significance: high
  • originality: high
  • clarity: good
  • formatting: excellent
  • grammar: good

Author:  Juraj Hasik  on 2025-01-06  [id 5084]

(in reply to Report 2 by Jutho Haegeman on 2024-09-04)
Category:
answer to question
correction

We thank the referee for the review and valuable suggestions, which have been incorporated into the resubmitted manuscript. Below, we address each point individually.

1) Around line104: it is perhaps worth mentioning that the charges for $Z_n$ are elements of $\mathbb{Z}_n$, because in those cases, the angle theta in the symmetry is restricted to take values that are integer multiples of $2\pi / N$. Also, I don't quite understand why the different symbol $Z_n$ and $\mathbb{Z}_n$ are used, but if the first one is supposed to refer to the multiplicative group, it is probably more correct to denote it as the cyclic group $C_n$.

Initially, we wanted to distinguish $C_n$ group (commonly denoted by $\mathbb{Z}_n$ in physics literature) and $\mathbb{Z}_n$ as set of allowed charges. Since the $C_n$ is isomorphic to $\mathbb{Z}_n$ (integers modulo n) the slopy notation is forgiving. We have revised this part and included suggested clarifications.

2) Around line 113: it is stated that boldface is used to underline the vector nature, but it seems that the SciPost style makes all math in boldface, so that also scalar values such as s, individual dimensions per sector $D_\rho$, and essentially all other math is typeset in boldface.

We added vector arrows to symbols describing (nested) tuples of charges and dimensions to avoid the ambiguity.

3) The final paragraph of section 2.1: can you elaborate on the the benefits of the lazy initialisation and the allocation of nonzero blocks only. Is it often the case that blocks which are allowed by the symmetry, turn out to be zero anyway? What happens if they receive nonzero data afterwards? Does this require to allocate a new tensor or to copy the data?

Adding a new block forces reallocation of the data. In the case of examples presented in the manuscript, i.e. iPEPS contractions via CTMRG and their optimization, the lazy initialization is marginal (performance-wise). Under tensor contraction paradigm via permutate -> reshape -> (block-sparse) matrix-multiply -> reshape, the resulting tensor has all blocks allowed by the outgoing legs present, and, filled with zero elements if the relevant blocks were not initialized in the original operands. Hence, through iterative algorithms one quickly ends up with tensors having all blocks allowed by leg structures filled. Still, the lazy initialization is a nice feature for contraction of general networks, which might not necessarily originate in physics domain or from iterative algorithms.

4) The code examples in lines 155-156 and 169-170 could perhaps use a bit more explanation, especially for non-Python users (although they might not be the typical audience of this paper). I was in particular wondering about what exactly the 'axes' argument in the 'tensordot' call specifies, and how the indices of the resulting tensor are sorted/defined. Another question is about the recording of the original structure of fusing legs: what happens if different fusion steps are concatenated, e.g. first 1 with 2 and 3 with 3, then the resulting 1 with the resulting 2. Are these intermediate steps recorded. Will 'unfuse_legs' take one step back, or immediately jump back to the original structure?

The entire history of fusion is stored (which is critical, as it allows resolving mismatches between fused spaces of two contracted tensors). unfuse_legs takes one step back. Currently, this implies data copy. Introducing views in unfuse operation is possible (and we plan for it), though the expected performance gain from doing so is only around a few percent. We expanded the description of the interface for fuse_legs and tensordot and a set of associated examples in the manuscript addressing the questions above.

5) Given that wall times are reported in seconds, would it be possible to provide some details on the hardware on which these simulations were ran?

We added a footnote at the beginning of the examples section, which gives details on the hardware.

6) More benchmarks can provide further interesting insight, and as listed in the report section, there are quite a few benchmarks that could be conceivable. I leave it to the authors to decide what they believe could be useful additions here.

We created a Github repository at https://github.com/yastn/yastn_benchmarks with DMRG comparison adopted by ITensor and TeNPy. This repo is now referenced in the main text. Ideally, besides the DMRG benchmark currently available, we would be interested in including TensorKit contraction benchmarks as well. We opt to leave out these benchmarks from the current manuscript, since they will become obsolete quite fast as further development continues. Instead, in the manuscript we focus on the structure of the tensors appearing in a few large applications. Those are implementation-independent, and show both the technical challenges related to symmetric implementations, as well as quantify the gains allowed by the symmetries. As such this should have a broader appeal. The structures of iPEPS tensors featuring in the examples are also part of the above benchmark repo. In particular, we observe that the available RAM memory is setting practical limits on feasible simulations and bond dimensions (which precludes using GPUs with limited memory in some applications).

7) I believe it would be good if the interplay/interdependency with peps-torch could be clarified. It is sometimes listed as an independent package (introduction, line 67, as well as conclusion, line 348), but at the same time seems closely tied to YASTN and is mentioned as part of it for doing the PEPS optimizations (e.g. conclusion, line 347).

This ambiguity really comes from the historical development of YASTN together with this manuscript. At the time when AD-optimization examples 3.1 and 3.2 were carried out, two algos were missing in YASTN: (i) generic CTMRG was not implemented in YASTN. Instead, we have used the generic CTMRG algorithm implemented in peps-torch with YASTN's symmetric tensor primitives. (ii) Gradient based optimizers are not part of YASTN. Hence, we have used optimizer implemented in peps-torch. The first point is no longer an issue. YASTN now comes with implementation of generic CTMRG. For the second point, however, the inclusion or tight integration of optimizers is questionable. Currently, running on PyTorch backend (and on JAX in the future), one can define and evaluate cost functions and their gradients within YASTN. That's sufficient for utilizing arbitrary gradient-based optimization algo, whether from SciPy, PyTorch, or even other packages. We will include a PyTorch-based optimizer in near future to provide sensible default, ideally sharing the custom one from peps-torch to avoid duplicity.

---

## Round 2 · Referee Report · Johannes Hauschild (Referee 1) · 2025-1-7

Report

The authors have addressed the weaknesses raised in the previous reports, and I recommend a publication as is.

Recommendation

Publish (easily meets expectations and criteria for this Journal; among top 50%)

---

## Round 2 · Author Response

Dear Editor,

We are pleased to resubmit our revised manuscript. In this revision, we have carefully addressed all the comments and suggestions provided by the reviewers and have incorporated their valuable feedback into the manuscript. We hope that the revised version meets the expectations of the reviewers and the editorial team.

Sincerely,
Juraj Hasik
on behalf of all authors

---

## Round 2 · List of Changes

• We created a new repository for performance benchmarks at https://github.com/yastn/yastn_benchmarks. It includes a DMRG benchmark used by both TenPy and ITensor, as well as iPEPS optimization benchmarks from examples 1 and 2 in the manuscript.
  • We now use overhead arrows for tuples of charges (or nested tuples for product groups) labeling charge sectors and tuples of sector bond dimensions.
  • We clarified the notation for cyclic group C_N (Z_N).
  • We expanded the descriptions of 'tensordot', 'fuse_legs', and 'unfuse_legs' functions, including syntax, leg ordering in tensordot results, and code examples demonstrating their usage.

---

## Editorial Decision

published